# RCT Rejection Sampling for Causal Estimation Evaluation

**Katherine A. Keith**                                    *kak5@williams.edu*
*Williams College*

**Sergey Feldman**                                    *sergey@allenai.org*
*Allen Institute for Artificial Intelligence*

**David Jurgens**                                    *jurgens@umich.edu*
*University of Michigan*

**Jonathan Bragg**                                    *jbragg@allenai.org*
*Allen Institute for Artificial Intelligence*

**Rohit Bhattacharya**                                    *rb17@williams.edu*
*Williams College*

**Reviewed on OpenReview:** *https://openreview.net/forum?id=F74ZZk5hPa*

## Abstract

Confounding is a significant obstacle to unbiased estimation of causal effects from observational data. For settings with high-dimensional covariates—such as text data, genomics, or the behavioral social sciences—researchers have proposed methods to adjust for confounding by adapting machine learning methods to the goal of causal estimation. However, empirical evaluation of these adjustment methods has been challenging and limited. In this work, we build on a promising empirical evaluation strategy that simplifies evaluation design and uses real data: subsampling randomized controlled trials (RCTs) to create confounded observational datasets while using the average causal effects from the RCTs as ground-truth. We contribute a new sampling algorithm, which we call *RCT rejection sampling*, and provide theoretical guarantees that causal identification holds in the observational data to allow for valid comparisons to the ground-truth RCT. Using synthetic data, we show our algorithm indeed results in low bias when oracle estimators are evaluated on the confounded samples, which is not always the case for a previously proposed algorithm. In addition to this identification result, we highlight several finite data considerations for evaluation designers who plan to use RCT rejection sampling on their own datasets. As a proof of concept, we implement an example evaluation pipeline and walk through these finite data considerations with a novel, real-world RCT—which we release publicly—consisting of approximately 70k observations and text data as high-dimensional covariates. Together, these contributions build towards a broader agenda of improved empirical evaluation for causal estimation.

## 1 Introduction

Across the empirical sciences, confounding is a significant obstacle to unbiased estimation of causal effects from observational data. Covariate adjustment on a relevant set of confounders aka *backdoor adjustment* (Pearl, 2009) is a popular technique for mitigating such confounding bias. In settings with only a few covariates, simple estimation strategies—e.g., parametric models or contingency tables—often suffice to compute the adjusted estimates. However, modern applications of causal inference have had to contend with thousands of covariates in fields like natural language processing (Keith et al., 2020; Feder et al., 2022), genetics (Stekhoven et al., 2012), or the behavioral social sciences (Li et al., 2016; Eckles & Bakshy, 2021). In

these high-dimensional scenarios, more sophisticated methods are needed and often involve machine learning. Recent approaches include non-parametric and semi-parametric estimators (Hill, 2011; Chernozhukov et al., 2018; Athey et al., 2018; Farrell et al., 2021; Bhattacharya et al., 2022), causally-informed covariate selection (Maathuis et al., 2009; Belloni et al., 2014; Shortreed & Ertefaie, 2017), proxy measurement and correction (Kuroki & Pearl, 2014; Wood-Doughty et al., 2018), and causal representation learning (Johansson et al., 2016; Shi et al., 2019; Veitch et al., 2020).

Despite all this recent work targeted at high-dimensional confounding, these methods have not been systematically and empirically benchmarked. Such evaluations are essential in determining which methods work well in practice and under what conditions. However, unlike supervised learning problems which have ground-truth labels available for evaluating predictive performance on a held-out test set, analogous causal estimation problems require ground-truth labels for counterfactual outcomes of an individual under multiple versions of the treatment, data that is generally impossible to measure (Holland, 1986).

A promising evaluation strategy is to directly subsample data from a randomized controlled trial (RCT) in a way that induces confounding. Causal effect estimates obtained using the confounded observational samples can then be compared against the ground-truth estimates from the RCT to assess the performance of different causal estimators. This idea has appeared in works like Hill (2011) and Zhang & Bareinboim (2021) and was recently formalized by Gentzel et al. (2021).

We contribute to this evaluation strategy — which we subsequently refer to as *RCT subsampling* — via theory that clarifies why and how RCT subsampling algorithms should be constrained in order to produce valid downstream empirical comparisons. In particular, we prove previous subsampling algorithms can produce observational samples from which the causal effect is provably not identified, which makes recovery of the RCT ground-truth impossible (even with infinite samples). To address this issue, we present a new RCT subsampling algorithm, which we call *RCT rejection sampling*, that appropriately constrains the subsampling such that the observed data distribution permits identification.

In addition to improving the theoretical foundations of RCT subsampling, we provide evaluation designers a scaffolding to apply the theory. We implement a proof of concept evaluation pipeline with a novel, real-world RCT dataset—which we release publicly—consisting of approximately 70k observations and text data as high-dimensional covariates. We highlight important finite data considerations: selecting an RCT dataset and examining when empirical evaluation is appropriate; empirically verifying a necessary precondition for RCT subsampling; specifying and diagnosing an appropriate confounding function using finite samples; applying baseline estimation models; and briefly speculate on additional challenges that could arise. For each of these considerations, we walk through specific approaches we take in our proof of concept pipeline.

In summary, our contributions are

- We provide a proof using existing results in causal graphical models showing that previous RCT subsampling procedures (e.g., Gentzel et al. (2021)) may draw observational data in a way that prevents non-parametric identification of the causal effect due to selection bias (§3.3).

- We propose a new subsampling algorithm, which we call *RCT rejection sampling*, that is theoretically guaranteed to produce an observational dataset where samples are drawn according to a distribution where the effect is identified via a backdoor functional (§3.4). Using three settings of synthetic data, we show our algorithm results in low bias, which is not always the case for a previous algorithm (§3.5).

- For evaluation designers who plan to use RCT rejection sampling for their own datasets, we highlight several finite data considerations and implement a proof of concept pipeline with a novel, real-world RCT dataset and application of baseline estimation models (§4).

- We release this novel, real-world RCT dataset of approximately 70k observations that has text as covariates (§4.1.1). We also release our code.[1]

These contributions build towards a more extensive future research agenda in empirical evaluation for causal estimation (§5).

---

[1]Code and data at `https://github.com/kakeith/rct_rejection_sampling`.

| Dataset | Eval. strategy | DoF | General Data avail. | DGP realism | Application to Backdoor Adjustment Covariates (num.) | Data public? |
|---|---|---|---|---|---|---|
| Simulation, normal assmpts. (D'Amour & Franks, 2021) | Synthetic | ✗ Many | ✓ High | ✗ Low | ✓ High (1000+) | ✓ Yes |
| IHDP-ACIC 2016 (Dorie et al., 2019) | Semi-synthetic | ✗ Many | ✓ High | ✗ Medium | ✗ Medium (58) | ✓ Yes |
| PeerRead theorems (Veitch et al., 2020) | Semi-synthetic | ✗ Many | ✓ High | ✗ Medium | ✓ High (Text vocab) | ✓ Yes |
| RCT repositories (Gentzel et al., 2021) | RCT subsampling | ✓ Few | ✓ High-RCTs | ✓ High | ✗ Low (1-2) | ✓ Yes |
| Job training (LaLonde, 1986) | COS | ✓ Few | ✗ Low | ✓ High | ✗ Low (4) | ✓ Yes |
| Facebook peer effects (Eckles & Bakshy, 2021) | COS | ✓ Few | ✗ Low | ✓ High | ✓ High (3700) | ✗ No |
| This work | RCT subsampling | ✓ Few | ✓ High-RCTs | ✓ High | ✓ High (Text vocab) | ✓ Yes |

Table 1: Select related work in empirical evaluation of causal estimators compared on general desiderata of: ✓ few degrees of freedom (DoF) for the evaluation designer, ✓ high data availability, and ✓ realistic data-generating processes (DGP). We also examine the accompanying datasets presented for evaluating backdoor adjustment. Here, we want a ✓ high number of covariates to make the evaluation non-trivial and ✓ public availability of the data for reuse and reproducibility.

# 2 Related Work in Empirical Evaluation of Causal Estimators

As we discussed briefly in Section 1, empirical evaluation of causal estimation methods for observational data is difficult but important.

We argue an evaluation strategy should in general (i) reduce the *evaluation designers' degrees of freedom* i.e. limit the number of choices researchers have to (inadvertently) pick an evaluation that favors their own method (Gentzel et al., 2019); (ii) has the necessary data (e.g., RCTs) available, (iii) ensure the data generating process (DGP) reflects the real world, and (iv) make the data publicly available for reuse and reproducibility. For applications of backdoor adjustment, we argue non-trivial evaluation should additionally (v) include a high number of covariates that could be used in the adjustment set. Table 1 compares select previous work (and our own) according to the above desiderata. We briefly discuss these and other related work, and make a qualitative argument for the RCT subsampling strategy we contribute to.

**Synthetic evaluations** are ones in which researchers specify the entire DGP, e.g., D'Amour & Franks (2021); Schmidt et al. (2022). This allows for infinite data availability, but is prone to encoding researcher preferences and can lead to over-simplification (or overly complex DGPs) compared to real-world observational scenarios.

**Semi-synthetic evaluations** use some real data but specify the rest of the synthetic DGP. This approach has been used in causal inference competitions (Dorie et al., 2019; Shimoni et al., 2018) and settings with text-data as confounding variables (Roberts et al., 2020; Veitch et al., 2020; Weld et al., 2022). Other semi-synthetic work fits generative models to real-world data (Neal et al., 2020; Parikh et al., 2022) or uses pre-trained language models to generate high-dimensional confounders from variables in a synthetic DGP (Wood-Doughty et al., 2021). Although more realistic than synthetic data, semi-synthetic DGPs can also make unrealistic assumptions; for example, Reisach et al. (2021) demonstrate this issue in the context of evaluating causal discovery algorithms.

**Constructed observational studies** (COSs) start with RCTs and then find non-experimental control samples that come from a similar population (LaLonde, 1986; Hill et al., 2004; Arceneaux et al., 2006; Shadish et al., 2008; Jaciw, 2016; Gordon et al., 2019; Eckles & Bakshy, 2021; Zeng et al., 2022; Gordon et al., 2022).[2] The advantage of COSs over (semi-)synthetic data is that they have few researcher degrees of freedom; however, non-experimental control groups often do not exist or do not come from similar-enough populations; see Dahabreh et al. (2022) for more details on identification from COSs.

**Subsampling RCTs** uses an RCT as ground-truth and then subsamples the RCT data to create a confounded observational dataset. For example, Zhang & Bareinboim (2021) subsample from the International Stroke Trial (IST) of roughly 20k patients to estimate the treatment effect of aspirin allocation. This strategy also appears in prior work (Hill, 2011; Kallus et al., 2018) and was recently formalized by Gentzel et al. (2021). While this approach is limited by the availability of RCTs and sampling decreases the number of units available to the estimation methods, it does not require the comparable non-experimental control group

---

[2]For example, LaLonde (1986) adjusts for four covariates—age, years of schooling, high school drop-out status, and race (Table 4, Footnote C in LaLonde)—in his non-experimental group in a study of the effects of job training on earnings.

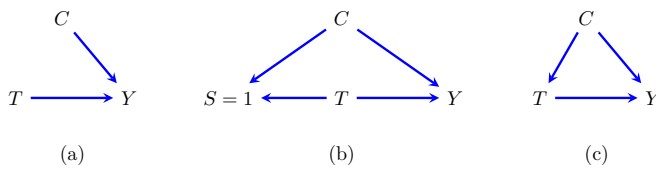

Figure 1: Causal DAGs (a) corresponding to an RCT; (b) representing a sampling procedure; (c) corresponding to an observational study where $C$ satisfies the backdoor criterion.

required by COSs, resulting in greater data availability. There are also fewer researcher degrees of freedom compared to synthetic or semi-synthetic approaches. Because of these tradeoffs, we believe this is one of the most promising strategies for empirically evaluating causal estimation methods and we build upon this strategy in the remainder of this work.

## 3    Subsampling from RCTs

We preface this section with a brief description of causal graphs, a prerequisite to understanding subsequent results. Then we provide identification and non-identification proofs for RCT subsampling algorithms and evidence from synthetic data.

### 3.1    Background: Causal graphical models

A causal model of a directed acyclic graph (causal DAG) $\mathcal{G}(V)$ can be viewed as the set of distributions induced by a system of structural equations: For each variable $V_i \in V$ there exists a structural equation $V_i \leftarrow f_i(\mathrm{pa}_i, \epsilon_i)$ (Pearl, 2009). This function maps the variable's parents' values—$\mathrm{pa}_i$ of $V_i$ in $\mathcal{G}(V)$—and an exogenous noise term[3], $\epsilon_i$, to values of $V_i$. The system of equations induces a joint distribution $P(V)$ that is Markov relative to the DAG $\mathcal{G}(V)$, i.e., $P(V = v) = \prod_{V_i \in V} P(V_i = v_i \mid \mathrm{Pa}_i = \mathrm{pa}_i)$. Independences in the distribution can be read off from $\mathcal{G}$ via the well-known d-separation criterion (Pearl, 2009). Interventions in the model are typically formalized using the do-operator (Pearl, 2009), where $Y \mid \mathrm{do}(T = t)$ denotes the value of an outcome $Y$ under an intervention that sets the treatment $T$ to value $t$.

Here, our causal estimand of interest is the average treatment effect (ATE) defined as,

$$\mathrm{ATE} \equiv \mathbb{E}[Y \mid \mathrm{do}(T = t)] - \mathbb{E}[Y \mid \mathrm{do}(T = t')], \tag{1}$$

where $t$ and $t'$ denote distinct values of $T$. A causal parameter is said to be *identified* if it can be expressed as a function of the observed data $P(V)$. Given a set of variables $Z \subset V$ that satisfy the *backdoor criterion* w.r.t $T$ and $Y$[4], the ATE is identified via the well-known *backdoor adjustment functional* (Pearl, 1995).

### 3.2    RCT subsampling: Setup and conditions

We now describe the specific setup and objectives of RCT subsampling.[5] We start with a dataset $D_{\mathrm{RCT}}$ consisting of $n$ iid draws from an RCT of pre-treatment covariates $C = \{C_1, \ldots, C_k\}$, treatment $T$, and outcome $Y$. Since this data is assumed to come from an RCT, the observed data distribution $P(C, T, Y)$

---

[3]Typically these noise terms are assumed to be mutually independent, but this assumption is not strictly necessary (Richardson & Robins, 2013). Our results are non-parametric in the sense that we do not require any distributional assumptions on the noise terms or the specific form (e.g., linearity) of the structural equations $f_i(\cdot)$.

[4]The set $Z$ satisfies the backdoor criterion if no variable in $Z$ is a causal descendant of $T$, and $Z$ blocks all backdoor paths between $T$ and $Y$, i.e., all paths of the form $T \leftarrow \cdots \rightarrow Y$.

[5]Here our target of interest is the ATE, though similar principles apply to other causal parameters like conditional average treatment effects.

is Markov relative to the causal DAG shown in Fig. 1(a) where $T \perp\!\!\!\perp C$. The goal of RCT subsampling is to construct an observational dataset $D_{\text{OBS}}$ that consists of $m \leq n$ iid draws from $D_{\text{RCT}}$ that satisfies the following conditions which enable appropriate evaluation of causal estimation methods:

(I) **Dependence induced.** $D_{\text{OBS}}$ consists of samples drawn according to a new distribution $P^*(C, T, Y)$ that satisfies the dependence relation $T \not\!\perp\!\!\!\perp C$.

(II) **ATE identified.** There exists a functional $g$ of the RCT distribution and a functional $h$ of the subsampled data distribution such that $\text{ATE} = g(P(C, T, Y)) = h(P^*(C, T, Y))$.

Here, (II) is an important identification pre-condition that ensures that it is possible, at least in theory, to compute estimates of the ATE from $D_{\text{OBS}}$ to match the ATE from $D_{\text{RCT}}$, the latter of which is treated as ground-truth in evaluation. From Fig. 1(a) it is clear that two sets of variables satisfy the backdoor criterion: the set $C$ and the empty set. Thus, the ATE is identified from the RCT distribution $P(C, T, Y)$ via the following two backdoor adjustment functionals,

$$\text{ATE} = \sum_c P(c) \times \big(\mathbb{E}[Y \mid t, c] - \mathbb{E}[Y \mid t', c]\big) \tag{2}$$

$$= \mathbb{E}[Y \mid t] - \mathbb{E}[Y \mid t']. \tag{3}$$

Thus, a subsampling algorithm satisfies (II) if there is a functional $h(P^*(C, T, Y))$ that is equal to equation 2 or equation 3.

For our purposes, we add the condition (I) so that estimation in the observational data does not reduce to equation 3. That is, we aim to produce samples according to a distribution $P^*$ such that some adjustment is in fact necessary to produce unbiased ATE estimates. We note that (I) by itself is not sufficient to guarantee this; RCT subsampling procedures also require that there exists at least one pre-treatment covariate correlated with the outcome, i.e., $\exists\ C_i \in C$ such that $C_i \not\!\perp\!\!\!\perp Y$ in $P(C, T, Y)$ (Gentzel et al., 2021). However, this condition is easily testable, and we implement these checks in our synthetic experiments and real-world proof of concept (§4.2).

We now show a theoretical gap in existing approaches to subsampling RCTs, and propose a new algorithm that is theoretically guaranteed to satisfy conditions (I) and (II).

## 3.3 Non-identification in prior work

We claim that prior work that proposes RCT subsampling can result in observational samples from which the causal effect is *not identified* non-parametrically unless additional constraints are placed on the subsampling process. We consider Algorithm 2 in Gentzel et al. (2021) which does not explicitly impose such constraints and can be summarized as follows. Let $S$ be a binary variable indicating selection into the observational data from $D_{\text{RCT}}$. A structural equation $S \leftarrow \mathbb{1}(T = \text{Bernoulli}(f(C)))$ is used to generate the selection variable, where $f$ is a function defined by the researcher and $\mathbb{1}$ corresponds to the indicator function. $D_{\text{OBS}}$ is created by retaining only samples from $D_{\text{RCT}}$ where $S = 1$. This results in $P^*(C, T, Y) = P(C, T, Y \mid S = 1)$ which is Markov relative to the causal DAG in Fig. 1(b). From this DAG, it is easy to check via d-separation that condition (I) is satisfied as $T \not\!\perp\!\!\!\perp C \mid S = 1$. However, the following proposition shows that condition (II) is not satisfied.

**Proposition 3.1.** *Given $n$ iid samples from a distribution $P$ that is Markov relative to Fig. 1(a), Algorithm 2 in Gentzel et al. (2021) draws samples according to a distribution $P^*$ such that condition (II) is not satisfied.*

We provide a proof in Appendix A.

The intuition behind the proof of Proposition 3.1 is as follows. Identification of the ATE relies on two pieces: the conditional mean of the outcome given treatment and covariates and the marginal distribution

---

**Algorithm 1** RCT rejection sampling

---

1: **Inputs:** $D_{\mathrm{RCT}}$ consisting of $n$ i.i.d. draws from $P(C, T, Y)$; $P^*(T|C)$, a function specified by evaluation designers; $M \geq \sup \frac{P^*(T|C)}{P(T)}$, a constant computed empirically

2: **Output:** $D_{\mathrm{OBS}}$, a subset of $D_{\mathrm{RCT}}$ constructed according to a distribution $P^*(C, T, Y)$ which satisfies conditions (I) and (II)

3:

4: **for** each unit $i \in D_{\mathrm{RCT}}$ **do**

5:   Sample $U_i$ uniform on $(0, 1)$

6:   **if** $U_i > \frac{P^*(T=t_i|C_i)}{\hat{P}(T=t_i)M}$ **then**

7:     Discard $i$

8:   **end**

9: **end for**

10: **Return:** $D_{\mathrm{OBS}} \leftarrow D_{\mathrm{RCT}} - \{\text{discarded units}\}$

---

of covariates. From Fig. 1(b), we have $\mathbb{E}[Y|T, C] = \mathbb{E}[Y|T, C, S = 1]$, but $P(C) \neq P(C|S = 1)$. Indeed this marginal distribution cannot be identified via any non-parametric functional of the subsampled distribution $P^*(C, T, Y)$ (Bareinboim & Tian, 2015). However, this non-identification result holds assuming that there is no additional knowledge/constraints on how $P^*$ is generated; in the next section we modify the sampling to place constraints on the generated distribution $P^*$ that mitigate this issue.

## 3.4   RCT rejection sampling

We propose Algorithm 1, which uses a rejection sampling procedure to subsample RCTs. Rejection sampling is useful when the target distribution is difficult to sample from but there exists a proposal distribution which is easier to sample from and the proposal distribution (times a constant) forms an "upper envelope" for the target distribution (Murphy, 2012, Chapter 23.2). Similar ideas on resampling data based on ratios of propensity scores appear in Thams et al. (2023) and Bhattacharya & Nabi (2022) in the context of testing independence constraints in post-intervention distributions. Though the rejection sampler also selects samples based on a function of $T$ and $C$, as in Fig. 1(b), we prove that additional constraints placed by the sampling strategy ensure identification holds in the new observed data distribution.

The intuition behind our algorithm is as follows. Sufficient constraints for maintaining identifiability of the ATE in $P^*(C, T, Y)$ via the functional in equation 2 are to ensure that $P^*(C) = P(C)$ and $P^*(Y \mid T, C) = P(Y \mid T, C)$.[6] When this holds, it follows that equation 2 is equivalent to the adjustment functional $h(P^*(C, T, Y)) = \sum_c P^*(c) \times (\mathbb{E}^*[Y \mid T = t, c] - \mathbb{E}^*[Y \mid T = t', c])$, where $\mathbb{E}^*$ denotes the expectation taken w.r.t $P^*(Y \mid T, C)$. To also satisfy (I), we propose resampling with weights that modify $P(T)$ to a new conditional distribution $P^*(T \mid C)$.

The considerations listed in the prior paragraph inform our choice of an acceptance probability of $\frac{1}{M} \times \frac{P^*(T|C)}{P(T)}$ in the rejection sampler, where $M$ is the usual upper bound on the likelihood ratio used in the rejection sampler, which in our case is $\frac{P^*(T|C)}{P(T)}$.[7] Here, $P^*(T \mid C)$ is a function specified by the evaluation designer that satisfies positivity ($\forall c, 0 < P^*(T \mid C = c) < 1$ almost surely), and is a non-trivial function of $C$ in the sense that $P^*(T \mid C) \neq P^*(T)$ for at least some values of $T$ and $C$.

**Theorem 3.2.** *Given $n$ iid samples from a distribution $P$ that is Markov relative to Fig. 1(a), a confounding function $P^*(T|C)$ satisfying positivity, and $M \geq \sup \frac{P^*(T|C)}{P(T)}$, the rejection sampler in Algorithm 1 draws samples from a distribution $P^*$, such that conditions (I) and (II) are satisfied.*

---

[6]One could also consider maintaining equality of just the conditional mean of $Y$ rather than the full conditional density.

[7]In practice, we approximate $M$ from $D_{\mathrm{RCT}}$ as $\frac{\max_{i \in \{1,\dots,n\}} P^*(T=t_i|C_i)}{\min_{i \in \{1,\dots,n\}} \hat{P}(T=t_i)}$.

| | Synthetic DGP Setting | Sampling Algorithm | Abs. Bias (std.) | Rel. Abs. Bias (std.) | CI Cov. |
|---|---|---|---|---|---|
| 1 | $|C| = 1$, $P(T = 1) = 0.3$ | Algorithm 2 from Gentzel et al. (2021) | 0.222 (0.010) | 0.089 (0.004) | 0.00 |
| | | RCT rejection sampling (This work) | **0.009 (0.007)** | **0.004 (0.003)** | 0.97 |
| 2 | $|C| = 1$, $P(T = 1) = 0.5$ | Algorithm 2 from Gentzel et al. (2021) | 0.009 (0.006) | 0.003 (0.003) | 0.98 |
| | | RCT rejection sampling | 0.007 (0.005) | 0.003 (0.002) | 0.98 |
| 3 | $|C| = 5$, Nonlinear | Algorithm 2 from Gentzel et al. (2021) | 0.252 (0.010) | 0.979 (0.037) | 0.00 |
| | | RCT rejection sampling | **0.012 (0.009)** | **0.046 (0.034)** | 0.98 |

Table 2: Absolute bias (abs. bias) between ATE from $D_{\text{RCT}}$ and the estimated ATE via backdoor adjustment on $D_{\text{OBS}}$ created by each sampling algorithm. We also report abs. bias relative to the RCT ATE (rel. abs. bias) and the mean and standard deviation (std.) across samples from 1000 random seeds. In the final column, we report the confidence interval coverage (CI Cov.)—the proportion of 1000 random seeds for which the 95% confidence interval contains the true (RCT) ATE. The DGPs for Settings 1-3 are given in Appendix C.

*Proof.* Rejection sampling generates samples from a target distribution $P^*(V_1, \ldots, V_k)$ by accepting samples from a proposal distribution $P(V_1, \ldots, V_K)$ with probability

$$\frac{1}{M} \times \frac{P^*(V_1, \ldots, V_k)}{P(V_1, \ldots, V_k)},$$

where $M$ is a finite upper bound on the likelihood ratio $P^*/P$ over the support of $V_1, \ldots, V_k$.

We start with samples from an RCT, so our proposal distribution factorizes according to the causal DAG in Fig. 1(a): $P(C, T, Y) = P(C) \times P(T) \times P(Y \mid T, C)$.

Our target distribution is one where $T \not\perp\!\!\!\perp C$, and factorizes as $P^*(C, T, Y) = P^*(C) \times P^*(T \mid C) \times P^*(Y \mid T, C)$, with additional constraints that $P^*(C) = P(C)$ and $P^*(Y \mid T, C) = P(Y \mid T, C)$. This establishes the likelihood ratio,

$$\frac{P^*(C, T, Y)}{P(C, T, Y)} = \frac{P(C) \times P^*(T \mid C) \times P(Y \mid T, C)}{P(C) \times P(T) \times P(Y \mid T, C)}$$
$$= \frac{P^*(T \mid C)}{P(T)},$$

and any choice of $M \geq \sup \frac{P^*(T|C)}{P(T)}$ used in the rejection sampler in Algorithm 1 produces samples from the desired distribution $P^*$, where the additional constraints satisfy the identification condition (II) and specification of $P^*(T|C)$ such that it truly depends on $C$ satisfies condition (I). □

Since $P^*$ satisfies $T \not\perp\!\!\!\perp C$ and yields identification via the usual adjustment functional obtained in a conditionally ignorable causal model, Algorithm 1 can be thought of as producing samples exhibiting confounding bias similar to the causal DAG in Fig. 1(c), despite the selection mechanism. A longer argument for this qualitative claim is in Appendix B.

We conclude this section by noting that similar to prior works on RCT subsampling algorithms, the subsampling strategy in Algorithm 1 only requires researchers to specify a single function, $P^*(T \mid C)$. Hence, our procedure satisfies our original desideratum of limited researcher degrees of freedom, while providing stronger theoretical guarantees for downstream empirical evaluation. However, specification of $P^*(T \mid C)$ may still be challenging when $C$ is high-dimensional. In Section 4.4, we discuss this finite data consideration and we use a proxy strategy for our proof of concept in which we have a low-dimensional confounding set $C$ along with a set of high-dimensional covariates $X$ that serve as proxies of this confounding set.

## 3.5 Evidence from synthetic data

Using synthetic DGPs for $D_{\text{RCT}}$, we produce $D_{\text{OBS}}$ using Algorithm 2 from Gentzel et al. (2021) and separately via our RCT rejection sampler. We then compute ATE estimates using equation 2 for $D_{\text{OBS}}$ and

compare it to the ground-truth estimates using equation 3 in $D_{\mathrm{RCT}}$. Section C in the appendix gives the full details of the data-generating processes (DGPs) for three settings. Briefly, the DGP in Setting 1 has a single confounding covariate $C$, sets $P(T = 1) = 0.3$, and has an interaction term $TC$ in the structural equation for $Y$. Setting 2 is the same as Setting 1 except we set $P(T = 1) = 0.5$. Setting 3 is a non-linear DGP with five covariates, $C_1, \ldots, C_5$. All methods are provided with the true adjustment set and functional form for the outcome regression, i.e., our experiments here use oracle estimators to validate the identification theory proposed in the previous subsection.

We construct 95% confidence intervals via bootstrapping and the percentile method Wasserman (2004) and report confidence interval coverage. After obtaining a sample $D_{\mathrm{OBS}}$ from the RCT via either RCT rejection sampling or Algorithm 2 from Gentzel et al., we resample $D_{\mathrm{OBS}}$ with replacement and calculate the ATE for that bootstrap sample. We repeat this for 1000 bootstrap samples and obtain a 95% confidence interval by taking the 2.5% and 97.5% points of the bootstrap distribution as the endpoints of the confidence interval. Across our 1000 random seeds of the synthetic DGPs, we measure the proportion of these confidence intervals that contain the true (RCT) ATE and report this metric as confidence interval coverage. See Appendix Section H for additional confidence interval plots.

Table 2 shows that our proposed RCT rejection sampler results in a reduction of absolute bias compared to Algorithm 2 from Gentzel et al. (2021) by a factor of over 24 for Setting 1 (0.22/0.009) and a factor of 21 in Setting 3 (0.252/0.012). For Setting 3, Gentzel et al.'s procedure results in almost a 100% increase in bias relative to the gold RCT ATE of $-0.26$. In Setting 2 where $P(T = 1) = 0.5$, the differences in absolute bias between the two algorithms is less pronounced.[8] In Settings 1 and 2, the confidence interval coverage for Gentzel et al.'s procedure is 0 whereas our RCT rejection sampling algorithm results in coverage of 0.97 and 0.98 for Settings 1 and 2 respectively, both slightly above the nominal 0.95 coverage. The results of the simulation are consistent with our theoretical findings that our algorithm permits identifiability under more general settings than prior work.

# 4 Finite Data Considerations and Proof of Concept

In the previous section, we provided theoretical guarantees for RCT rejection sampling and confirmed the algorithm results in low bias on synthetic data. In this section, we demonstrate how to put this proposed theory into practice and highlight considerations when working with finite real-world data. Our goal is to surface questions that must be asked and answered in creating useful and high-quality causal evaluation.

We also describe our specific approach towards each consideration as we create a proof of concept pipeline for empirical evaluation of high-dimensional backdoor adjustment methods. For this proof of concept, we use a large-scale, real-world RCT dataset with text as covariates. Although our approaches are specific to our proof of concept dataset, we believe other evaluation designers will benefit from a real-world example of how to put the theory and considerations into practice.

## 4.1 Considerations prior to using a specific RCT dataset

A necessary component for RCT subsampling is obtaining a real-world RCT dataset. This ensures a more realistic data generating processes compared to synthetic or semi-synthetic approaches (see Table 1). As Gentzel et al. (2021) note, there are many RCT repositories from a variety of disciplines from which evaluation designers could gather data. However, Gentzel et al. find many of these existing datasets only have one or two covariates that satisfy $C \not\perp\!\!\!\perp Y$ (see Consideration #1 below).

As we briefly mentioned in Section 1, for settings with just a few covariates one can often use simple estimation strategies with theoretical guarantees—e.g., parametric models or contingency tables—and empirical evaluation may not be particularly informative in this setting. Along these lines, we recommend that evalu-

---

[8]We note Gentzel et al. (2021) primarily focus on the setting for which $P(T = 1) = P(T = 0) = 0.5$. However, their approach does not seem to generalize well outside of this setting, theoretically and empirically.

ation designers first ask themselves, *Is empirical evaluation of causal estimators appropriate and necessary for this setting?* Not all settings are in need of empirical evaluation.

A potentially untapped resource for RCT rejection sampling data is A/B tests from large online platforms. Other work, e.g., Eckles & Bakshy (2021), have used these types of experiments for constructed observational studies and we use such a dataset in our proof of concept. The large scale of these experiments can be advantageous since RCT subsampling reduces the number of units in the observational dataset by roughly half. Further, they often contain rich metadata and many covariates, which can be used to induce confounding in a way that emulates a high-dimensional setting.

### 4.1.1 Proof of concept approach

For our proof of concept, we choose a setting for which empirical evaluation is appropriate and needed: high-dimensional backdoor adjustment. Our high-dimensional covariates are the thousands of vocabulary words from text data, an application area that has generated a large amount of interest from applied practitioners, see Keith et al. (2020); Feder et al. (2022).

We use and publicly release a large, novel, real-world RCT (approximately 70k observations) that was run on an online scholarly search engine.[9] Users arrive on a webpage that hosts metadata about a single academic paper and proceed to interact with this page. The RCT's randomized binary treatment is swapping the ordering of two buttons—a PDF reader and a new "enhanced reader". We set $T = 1$ as the setting where the "enhanced reader" is displayed first. The outcome of interest is a user clicking ($Y = 1$) or not clicking ($Y = 0$) on the enhanced reader button. The former action transports the user to a different webpage that provides a more interactive view of the publication. The RCT suggests that the treatment has a positive causal effect with an ATE of 0.113 computed using a simple difference of conditional means in the treated and untreated populations. See Appendix D for more details about the RCT.

## 4.2 Consideration #1: Checking necessary precondition $C \not\perp\!\!\!\perp Y$

As we mentioned in Section 3, a necessary precondition for RCT subsampling in general is the existence of a causal edge between $C$ and $Y$, implying $C \not\perp\!\!\!\perp Y$. The relationship between $C$ and $Y$ is naturally occurring (not modified by evaluation designers) and the amount of confounding induced by sampling is, in part, contingent on this relationship (Gentzel et al., 2021). One can empirically check $C \not\perp\!\!\!\perp Y$ via independence tests, e.g., evaluating the odds ratio when both $C$ and $Y$ are binary variables. If there do not exist covariates, $C$ such that $C \not\perp\!\!\!\perp Y$, one cannot move forward in the evaluation pipeline using RCT subsampling.

### 4.2.1 Proof of concept approach

For our proof of concept, we use a subpopulation strategy to ensure the precondition $C \not\perp\!\!\!\perp Y$ is satisfied. We choose a single interpretable covariate to induce confounding: the field of study of manuscripts. Since $C \perp\!\!\!\perp Y$ if and only if the odds ratio between $C$ and $Y$ is 1, we choose subpopulations of the full RCT that have high a odds ratio between a subset of the categorical field of study variable and the outcome $Y$. Specifically, we choose $C$ to be a binary covariate representing one of two fields of study; for *Subpopulation A*, the field is either Physics or Medicine. In Appendix G, we implement the evaluation pipeline for an additional subpopulation with $C$ chosen as the articles with Engineering or Business as the field of study. Substantively, one can interpret this high odds ratio as natural differences in click rates from users viewing articles from different fields of study.

We combine this subpopulation strategy with a proxy strategy in the next section to ensure that the estimation procedures only have access to high-dimensional covariates instead of our low-dimensional $C$. This has

---

[9]The RCT was conducted on the Allen Institute for Artificial Intelligence's Semantic Scholar platform `https://www.semanticscholar.org/`. Owners of the website conducted this experiment and gave us permission to use and release this data.

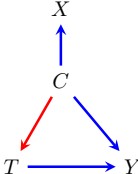

| RCT Dataset | $C$ categories | n | RCT ATE | $OR(C, Y)$ |
|---|---|---|---|---|
| Full | - | 69,675 | 0.113 | - |
| Subpopulation A | Physics, Medicine | 4,379 | 0.096 | 1.8 |

Figure 2: Proof of concept approach. **Left figure.** Causal DAG for the proxy strategy. The blue edges are confirmed to empirically exist in the finite dataset. The red edge is selected by the evaluation designers via $P^*(T|C)$. **Right table.** RCT dataset descriptive statistics including the number of units in the population/subpopulation ($n$) and the odds ratio, $OR(C, Y)$.

the benefit of simplifying the process of specifying $P^*(T \mid C)$ while still ensuring that downstream modeling must still contend with high-dimensional adjustment.

## 4.3 Consideration #2: Specification of $P^*(T|C)$

Evaluation designers using RCT rejection sampling have one degree of freedom: specification of $P^*(T|C)$. We describe one specific approach in our proof of concept to choose $P^*(T|C)$, but we anticipate evaluation designers using RCT rejection sampling to create large empirical benchmarks may want to include many different parameterizations of $P^*(T|C)$ to evaluate empirical performance of methods under numerous settings. Consideration #3 describes approaches to diagnosing the choice of $P^*(T|C)$ for a specific finite dataset.

### 4.3.1 Proof of concept approach

**Proxy strategy.** In Section 3, we briefly mentioned that specifying a suitable confounding function $P^*(T \mid C)$ may be difficult when $C$ is high-dimensional. A key property of our RCT is that it has high-dimensional text data, $X$, that is a proxy (with almost perfect predictive accuracy) for low-dimensional structured metadata—categories of scientific articles, e.g., Physics or Medicine. We use this structured metadata as the covariates $C$ in our RCT rejection sampler, but provide the causal estimation methods only $X, Y$ and $T$. Note that as evaluation designers, we still have access to $C$ to run diagnostics. This proxy strategy helps simplify the specification of $P^*(T \mid C)$ we use in the rejection sampler and avoids direct specification of a function involving high-dimensional covariates. Others have used similar proxy strategies for text in semi-synthetic evaluations, e.g., Roberts et al. (2020); Veitch et al. (2020). Such a technique may also be applied in other RCTs, e.g., healthcare studies where one or two important biomarkers serve as low-dimensional confounding variables, and the larger electronic health record data serves as the proxy $X$.

**Using $X$.** For each document $i$, the high-dimensional covariates $X_i$ is a bag-of-words representation of the document's concatenated title and abstract given a 2000-unigram vocabulary. A new vocabulary is created for each RCT subpopulation; see Appendix E for details. We check that there is high predictive accuracy of $P(C|X)$ to ensure the plausibility of causal estimation models only having access to $X$.[10] To measure this predictive accuracy, we model $P(C|X)$ with a logistic regression[11] classifier. Averaged across held-out test folds, the F1 score is 0.98 and the average precision is 0.99 for Subpopulation A (Physics, Medicine).

**Specifying $P^*(T|C)$.** Since $C$ is binary in our proof of concept pipeline, we choose a simple interpretable piece-wise function,

$$P^*(T_i = 1|C_i) = \begin{cases} \zeta_0 \text{ if } C_i = 0 \\ \zeta_1 \text{ if } C_i = 1 \end{cases} \tag{4}$$

for each document $i$, where $0 < \zeta_0 < 1$ and $0 < \zeta_1 < 1$ are parameters chosen by the evaluation designers. We choose $\zeta_0, \zeta_1$ for the remainder of the proof of concept pipeline via the diagnostics in Consideration #3.

---

[10]We leave to future work correcting for measurement error with noisy proxies $X$; see Wood-Doughty et al. (2018).

[11]Using scikit-learn Pedregosa et al. (2011) and an elasticnet penalty, L1 ratio 0.1, class-weight balanced, and SAGA solver.

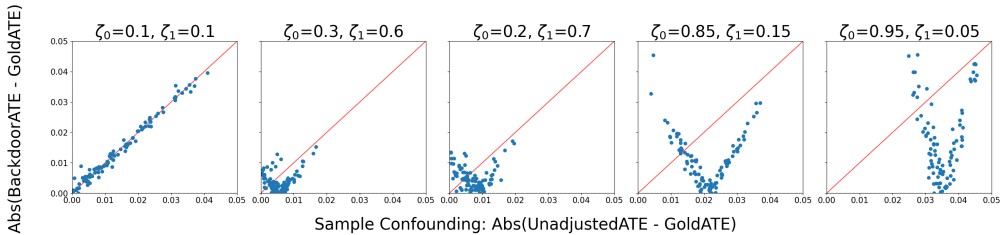

Figure 3: **Diagnostic plots for proof of concept pipeline.** For Subpopulation A data, each plot is the parameterization of $P^*(T|C)$ in Equation 4, which is specified by the evaluation designer. Each blue circle is a different random seed (100 seeds total per plot/parameterization).

## 4.4 Consideration #3: Diagnostics for $P^*(T|C)$

We recommend running diagnostics on the sampling procedure because, although Theorem 3.2 proves that our RCT rejection sampling permits identification, this is an asymptotic statement, and the sampler can produce finite samples where confounding bias is induced, but difficult to correct for.

### 4.4.1 Proof of concept approach

For our proof of concept pipeline, we step through the following diagnostics on the sampling procedure. Recall the notation from Section 3, $D_{\text{RCT}}$ is our RCT dataset (here, from Subpopulation A) and $D_{\text{OBS}}$ is the resulting observational dataset after RCT rejection sampling. First, we check empirically that overlap is satisfied for $C$ in $D_{\text{OBS}}$, i.e., $0 < \hat{P}(T = 1|C = c) < 1$ for all $c$.

Second, in Figure 3 we compare the amount of confounding induced to the error in the oracle adjustment for different $\zeta_0, \zeta_1$ in Equation 4 across 100 random seeds (blue dots). On the y-axis, we plot the absolute difference between the ATE for $D_{\text{RCT}}$ (GoldATE) and exact backdoor estimates[12] obtained from using the oracle adjustment set $C$. On the x-axis, we plot the absolute difference between the ATE on $D_{\text{RCT}}$ (GoldATE) and the unadjusted naive estimate on $D_{\text{OBS}}$. In general, we want more samples to fall below the $y = x$ line (shown in red) since this means that more confounding was induced than there is error in estimation. Samples above the $y = x$ have more error from the sampling process than the amount of confounding induced, and thus are not useful in benchmarking methods that adjust for confounding. Of the settings in Figure 3, $\zeta_0 = 0.85$ and $\zeta_1 = 0.15$ had the best proportion of sampled datasets lying below the red line, so we choose these parameters for our proof of concept pipeline. We leave to future work providing more guidance on choosing settings of $P^*(T|C)$ for a comprehensive benchmark.

## 4.5 Consideration #4: Modeling

The primary goal of this work is to create clear steps to follow during the evaluation design phase. Although this stage precedes a thorough modeling effort, we recommend that one runs baseline models to check for potential issues.

### 4.5.1 Proof of concept approach

As a proof of concept, we apply baseline causal estimation models to the resulting $D_{\text{OBS}}$ datasets after RCT rejection sampling (with many random seeds); as we mention above. We implement[13] commonly-used causal estimation methods via two steps: (1) fitting base learners and (2) using causal estimators that combine

---

[12]The exact backdoor equation is tractable because $T, C$ and $Y$ are all binary.

[13]We attempted to use the `EconML` package Microsoft Research (2019), but at of the time of our experiments it did not support using sparse matrices, which is required for our high-dimensional datasets.

| Prediction Ave. Prec. ($\uparrow$ better) | $\hat{g}(x)$ train | inference | $\hat{Q}_{T_0}(x)$ train | inference | $\hat{Q}_{T_1}(x)$ train | inference | $\hat{Q}_X(x)$ train | inference |
|---|---|---|---|---|---|---|---|---|
| linear | 0.89 (0.07) | 0.59 (0.02) | 0.63 (0.32) | 0.03 (0.01) | 0.85 (0.17) | 0.13 (0.03) | 0.71 (0.2) | 0.06 (0.01) |
| catboost (nonlinear) | 0.97 (0.0) | 0.60 (0.02) | 1.0 (0.0) | 0.03 (0.01) | 0.99 (0.01) | 0.13 (0.02) | 0.98 (0.01) | 0.05 (0.01) |

| Causal Rel. Abs. Error ($\downarrow$ better) | Unadjusted (baseline) | Backdoor $C$ (oracle) | $\hat{\tau}_Q$ | $\hat{\tau}_{\text{IPTW}}$ | $\hat{\tau}_{\text{AIPTW}}$ | $\hat{\tau}_{\text{DML}}$ |
|---|---|---|---|---|---|---|
| linear | 0.21 (0.08) | 0.12 (0.09) | 1.46 (1.04) | 0.47 (0.16) | 1.6 (0.66) | 1.91 (0.9) |
| catboost (nonlinear) | 0.21 (0.08) | 0.12 (0.09) | 0.24 (0.1) | 0.14 (0.11) | 0.11 (0.1) | 0.13 (0.1) |

Table 3: Modeling results for subpopulation A. **Top:** Predictive models' average precision (ave. prec.) for training (yellow) and inference (green) data splits. **Bottom:** Causal estimation models' relative absolute error (rel. abs. error) between the models' estimated ATE and the RCT ATE. Here, darker shades of red indicate worse causal estimates. Baselines, unadjusted conditional mean on the samples (unadjusted) and the backdoor adjustment with the oracle $C$ (backdoor C), are uncolored. We use two baselearner settings: linear and catboost (nonlinear). We report both the average and standard deviation (in parentheses) over 100 random seeds during sampling. All settings use $P^*(T|C)$ in equation 4 parameterized by $\zeta_0 = 0.85, \zeta_1 = 0.15$.

the base learners via plug-in principles or second-stage regression. We took care to ensure we use the *same* pre-trained base learners—same functional form and learned weights—as inputs into any appropriate causal estimator.

**Base learners.** We implement base learners for:[14]

$$Q_{T_0}(x) := \mathbb{E}[Y|T=0, X=x] \tag{5}$$
$$Q_{T_1}(x) := \mathbb{E}[Y|T=1, X=x] \tag{6}$$
$$g(x) := P(T=1|X=x) \tag{7}$$
$$Q_X(x) := \mathbb{E}[Y|X=x] \tag{8}$$

In our application both $T$ and $Y$ are binary variables, so we use an ensemble of gradient boosted decision trees (catboost) [15] and logistic regression[16] for our base learners. We fit our models using cross-fitting (Hansen, 2000; Newey & Robins, 2018) and cross-validation; see Appendix F for more details.

**Causal estimators.** After training base learners with cross-fitting, we implement the following plug-in causal estimators: backdoor adjustment (outcome regression) (Q), inverse propensity of treatment weighting (IPTW), and augmented inverse propensity of treatment weighting (AIPTW) (Robins et al., 1994). We also use DoubleML (Chernozhukov et al., 2018) which applies an ordinary least squares on residuals from the base learners. See Appendix F for exact estimation equations.

**Modeling results.** Table 3 shows results for Subpopulation A. Since both $T$ and $Y$ are binary, we report average precision (AP) for the base learners on both the training and inference folds; this metric is only calculated for observed (not counterfactual) observations. We also report the relative absolute error (RAE) between estimators on $D_{\text{OBS}}$ and the ATE for $D_{\text{RCT}}$. Comparing predictive base learners, the propensity score model, $g(x)$, has much higher AP on inference folds than models that involve the outcome. As we previously mentioned, RCT subsampling allows us to set the relationship between $X$ (via proxy for $C$) and $T$ but not $X$ and $Y$ so the low AP for outcome models could reflect the difficulty in estimating this "natural" relationship between $X$ and $Y$. See Section 4.6.1 for additional discussion on low average precision for the outcome models (Q).

For causal estimators, we see that the doubly robust estimator AIPTW using catboost has the lowest estimation error—on par with estimates obtained using the oracle backdoor adjustment set $C$. It appears the doubly robust estimators using linear models do not recover from the poor predictive performance of the outcome models and IPTW is better in this setting. Though seemingly counterintuitive that the linear and catboost models have similar predictive performance but large differences in the causal estimation error,

---

[14] Note for a binary outcome $Y$, we can rewrite the above equations with probabilities, as $\mathbb{E}[Y \mid \cdot] = P(Y=1 \mid \cdot)$.

[15] Using CatBoost (Dorogush et al., 2018) with default parameters and without cross-validation.

[16] Using scikit-learn (Pedregosa et al., 2011) and an elasticnet penalty, L1 ratio 0.1, balanced class weights, and SAGA solver. We tune the regularization parameter $C$ via cross-validation over the set $C \in 1e^{-4}, 1e^{-3}, 1e^{-2}, 1e^{-1}, 1e^0, 1e^1$.

this discrepancy between predictive performance and causal estimation error is consistent with theoretical results on fitting nuisance models in causal inference (Tsiatis, 2007; Shortreed & Ertefaie, 2017) and empirical results from semi-synthetic evaluation (Shi et al., 2019; Wood-Doughty et al., 2021). Although we used best practices from machine learning to fit our base learners, these results suggest future work is needed to adapt machine learning practices to the goals of causal estimation.

## 4.6 Consideration #5: Additional finite data challenges

The broader purpose of this line of empirical evaluation is to understand the real-world settings for which certain estimation approaches are successful or unsuccessful. We believe bridging the gap between theory that holds asymptotically and finite data is important for drawing valid causal inference, but evaluation designers might encounter unforeseen challenges particular to finite data that need to be examined carefully.

### 4.6.1 Proof of concept approach

In our proof of concept pipeline, we hypothesize the outcome models have very low average precision (Table 3) because of finite data issues with class imbalance. In particular, for Subpopulation A, 82% of our data is $C = 1$ and $\mathbb{E}[Y] = 0.07$ so there are few examples to learn from in the smallest category ($C = 0$, $Y = 1$): only 34 documents. This shows that even with our RCT that has a relatively large size (roughly 4k units) compared to other real-world RCTs, there are challenges with having sufficient support.

## 5 Discussion and Future Work

Unlike predictive evaluation, empirical evaluation for causal estimation is challenging and still at a nascent stage. In this work, we argue that one of the most promising paths forward is to use a RCT subsampling strategy, and to this line of work, we contribute an RCT rejection sampler with theoretical guarantees. We showed the utility of this algorithm in a proof of concept pipeline with a novel, real-world RCT to empirically evaluate high-dimensional backdoor adjustment methods.

Of course, there are critics of emprical evaluation. Hernán (2019) pejoratively compared a competition for causal estimation to evaluating "spherical cows in a vacuum" and claimed this discounted necessary subject-matter expertise. Even in the machine learning community, researchers warn against "mindless bake-offs" of methods (Langley, 2011), and in some cases the creation of benchmarks has led to the community overfitting to benchmarks, e.g., Recht et al. (2019). However, in the absence of theory or when theoretical assumptions do not match reality, we see empirical evaluation as a necessary, but not exclusive, part of the broader field of causal inference.

A fruitful future direction is for evaluation designers to use our RCT rejection sampler to create comprehensive benchmarks for various phenomena of interest: not only high-dimensional confounding but also heterogeneous treatment effects, unmeasured confounding, missing data, etc. This would involve gathering more RCTs and establishing interesting ways to set $P^*(T|C)$. Our proof of concept evaluation pipeline demonstrated the utility of RCT subsampling but there were many avenues we chose not to pursue such as: measurement error, causal null hypothesis tests, or moving to more sophisticated natural language processing approaches beyond bag-of-words, e.g., the CausalBERT model (Veitch et al., 2020).

In another direction, applied practitioners need guidance on which causal estimation method to use given their specific observational data. Although other work has attempted to link observational data to experimental data (real or synthetic) in which the ground-truth is known (Neal et al., 2020; Kallus et al., 2018), we believe RCT subsampling could help with meta analyses of which combination of techniques work best under which settings. Overall, we see this work as contributing to a much larger research agenda on empirical evaluation for causal estimation.

## Broader Impact Statement

We conducted this research with ethical due diligence. Our real-world RCT dataset was implemented by owners of the online platform and in full compliance with the platform's user agreement. The platform owners gave us explicit permission to use and access this dataset. Our dataset contains paper titles and abstracts, which are already publicly available from many sources, and we have removed any potentially personally identifiable information from the dataset, e.g., author names, user ids, user IP addresses, or session ids. By releasing this data, we do not anticipate any harm to authors or users.

Like any technological innovation, our proposed RCT rejection sampling algorithm and evaluation pipeline have the potential for dual use—to both benefit or harm society depending on the actions of the humans using the technology. We anticipate there could be substantial societal benefit from more accurate estimation of causal effects of treatments in the medical or public policy spheres. However, other applications of causal inference could potentially harm society by controlling or manipulating individuals. Despite these tradeoffs in downstream applications, we feel strongly this paper's contributions will result in net overall benefit to the research community and society at large.

## Author Contributions

KK conceived the original idea of the project and managed the project. RB contributed the ideas behind Algorithm 1 as well as the proofs in Section 3 and the Appendix. RB and KK implemented the synthetic experiments in Section 3. KK gathered and cleaned the data for the proof of concept pipeline in Section 4. KK and SF implemented the proof of concept empirical pipeline in Section 4. KK and RB wrote the first draft of the manuscript. KK, SF, DJ, JB, and RB guided the research ideas and experiments and edited the manuscript.

## Acknowledgments

The authors gratefully thank David Jensen, Amanda Gentzel, Purva Pruthi, Doug Downey, Brandon Stewart, Zach Wood-Doughty and Jacob Eisenstein for comments on earlier drafts of this manuscript. The authors also thank anonymous reviewers from ICML and TMLR for helpful comments. Special thanks to the Semantic Scholar team at the Allen Institute for Artificial Intelligence for help gathering the real-world RCT dataset.

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

# A    Proof of Proposition 3.1

*Proof.* Since the selection variable $S$ is created using a structural equation dependent on $T$ and $C$ and samples are retained only when $S = 1$, the distribution of the selected samples $P^*(C, T, Y) = P(C, T, Y \mid S = 1)$ and is Markov relative to Fig. 1(b). In the absence of additional constraints on $P^*$ (e.g., linearity assumptions) or the relation between $P$ and $P^*$, it is known that the ATE is non-parametrically identified in the presence of selection bias if and only if no variable that is a causal ancestor of the outcome in a graph where one deletes the treatment variable is also a causal ancestor of the selection variable (Bareinboim & Tian, 2015, Theorem 2). In this case, if we delete $T$ from Fig. 1(b), $C$ remains a causal ancestor of $Y$ and $S$. Hence, the effect is not identified via any non-parametric functional of the observed data and condition (II) cannot be satisfied.                                                                                                      □

# B    Viewing Algorithm 1 as drawing from a conditionally ignorable causal model

A causal model of a DAG $\mathcal{G}(V)$ can be interpreted as (i) a set of statistical distributions $P(V)$ that factorize according to $\mathcal{G}$: $P(V) = \prod_{V_i \in V} P(V_i \mid \mathrm{Pa}_i)$; and (ii) a set of post intervention distributions given by the g-formula aka truncated factorization (Robins, 1986; Spirtes et al., 2000; Pearl, 2009): for every $A \subset V$, we have $P(V \setminus A \mid \mathrm{do}(A = a)) = \prod_{V_i \in V \setminus A} P(V_i \mid \mathrm{Pa}_i) \mid_{A=a}$.

If it were possible to recollect data through a conditionally randomized experiment where treatment is assigned with probability $P^*(T \mid C)$ instead of $P(T)$, the observed data distribution over $C, T, Y$ would factorize according to the standard conditionally ignorable model shown in Fig. 1(c). That is, the distribution factorizes as $P^*(C, T, Y) = P(C) \times P^*(T \mid C) \times P(Y \mid T, C)$ and implies no independence constraints on the observed data. The distribution of samples $P^*(C, T, Y)$ output by the rejection sampler in Algorithm 1 also implies no independence constraints on $C, T, Y$ and thus has the exact same factorization. This establishes statistical equivalence of the conditionally ignorable model and the one obtained via our rejection sampler.

However, statistical equivalence alone is insufficient, as it is statistically equivalent to any complete DAG on the variables $C, T, Y$ that imply no independence constraints on the observed data. However, it is easy to see that all post-intervention distributions obtained via truncated factorization in the conditionally ignorable model also have the same identifying functionals in the model obtained through the rejection sampler due to the extra constraints imposed in Algorithm 1. For example, $P^*(T, Y \mid \mathrm{do}(C = c)) = P^*(T \mid C = c) \times P(Y \mid T, C)$ and $P^*(C, Y \mid \mathrm{do}(T = t)) = P(C) \times P(Y \mid T = t, C)$ in both the conditionally ignorable model and the one obtained via rejection sampling. Since the distribution obtained from the rejection sampler is indistinguishable from a conditionally ignorable model both in terms of the statistical model and all post-intervention distributions, applying Algorithm 1 can be viewed as essentially drawing samples from a standard conditionally ignorable model. That is, despite the selection mechanism, the resulting distribution is indistinguishable from Figure 1(c), the regular causal DAG model we use to represent observed confounding.

# C    Synthetic DGPs to evaluate sampling algorithms

We use the following data-generating process (DGP) to create synthetic data with 100k units for which the true ATE is equal to 2.5. We call this **Setting 1**:

$$C \sim \mathrm{Binomial}(0.5)$$
$$T \sim \mathrm{Binomial}(0.3)$$
$$Y \sim 0.5C + 1.5T + 2TC + \mathrm{Normal}(0, 1)$$

We set the researcher-specified confounding function $P^*(T = 1|C) = \sigma(-1 + 2.5C)$, where $\sigma$ is the logistic (expit) function, for RCT Rejection sampling and the same function as $f$ in Gentzel et al. (2021)'s Algorithm 2.

**Setting 2** has the same DGP as Setting 1 but we change $T \sim \text{Binomial}(0.5)$. For Settings 1 and 2, we provide the backdoor adjustment estimator with the true adjustment set, such that it adjusts for both $C$ and the interaction term, $TC$.

**Setting 3** involves more covariates that are combined in non-linear ways:

$$
\begin{aligned}
C_1 &\sim \text{Binomial}(0.5) \\
C_2 &\sim C_1 - \text{Uniform}(-0.5, 1) \\
C_3 &\sim \text{Normal}(0, 1) \\
C_4 &\sim \text{Normal}(0, 1) \\
C_5 &\sim C_3 + C_4 + \text{Normal}(0, 1) \\
T &\sim \text{Binomial}(0.3) \\
Y &\sim 0.5C_4 + 2TC_1C_2 - 1.5T + C_2C_3 + C_5 + \text{Normal}(0, 1)
\end{aligned}
$$

In this setting, we set $P^*(T = 1|C) = \sigma(0.5C_1 + -0.7C_2 + 1.2C_3 + 1.5C_4 + -1.2C_5 + 0.5C_1C_2)$. We provide the parametric backdoor estimator with the true adjustment set: $C_4, TC_1C_2, C_2C_3, C_5$.

See Table 4 for the ATEs for each of the three settings.

| DGP Setting | RCT ATE |
|---|---|
| 1 - Linear, $P(T = 1) = 0.3$ | 2.48 |
| 2 - Linear, $P(T = 1) = 0.5$ | 2.49 |
| 3 - Nonlinear, 5 $C$'s | -0.26 |

Table 4: RCT ATEs for the synthetic DGPs.

# D   RCT dataset details

We expand on the details of the RCT described in Section 4.1.1. The RCT was conducted on the Allen Institute for Artificial Intelligence's Semantic Scholar[17] platform.

The experiment ran for 25 days from 2022-06-04 to 2022-06-28 and resulted in 53,281 unique users arriving on 50,833 unique paper pages. This is after filtering to users who are "active", meaning prior to the experiment they clicked somewhere on the website at least once. Treatment is randomized for the combination of a unique user's browser plus user's device. We then post-process to recognize logged-in users across devices/browsers and remove them from the results if they switched treatments. The outcome of interest is if a user clicks on the "enhanced reader" button at least once during a session ($Y = 1$).

Intuitively, a positive causal effect is expected since we expect advertising a feature of the website to result in increased click rate. The final ATE was 0.113 as computed by a simple difference in conditional means $\mathbb{E}[Y \mid T = 1] - \mathbb{E}[Y \mid T = 0]$.

# E   Creating vocabulary for $X$

A new vocabulary is created for each RCT subpopulation. To create each vocabulary, we use binary indicators, remove stopwords, remove numbers and strip accents. Words must occur in at least 5 documents. We ignore terms that have a document frequency strictly lower than 10%.

---

[17]https://www.semanticscholar.org/

# F Modeling

**Base learners.** Base learners $Q_{T_0}$ and $Q_{T_1}$ in Equations 5 and 6 respectively are fit as described by Künzel et al. (2019) as "T-learners" (*not* "S-learners"), i.e. we take all samples for which we have observed $T = 0$ and then regress $X$ on $Y$ to get a trained model for $Q_{T_0}$ and likewise for units with observed $T = 1$ and $Q_{T_1}$. In preliminary experiments, we used the "S-learner" but found high-dimensional $X$ dominated $T$ and there were no differences learned between observed and counterfactual $T$ settings.

**Cross-fitting with cross-validation.** We fit our models using *cross-fitting* (Newey & Robins, 2018) which is also called sample-splitting (Hansen, 2000). Here, we divide the data into $K$ folds. For each inference fold $j$, the other $K - 1$ folds (shorthand $-j$) are used as the training set to fit the base learners—e.g., $\hat{Q}_{T_0}^{-j}$ or $\hat{g}^{-j}$—where the superscript here indicates the data the model is fit on. The single hyperparameter for logistic regression is selected via cross-validation, where the training set is again split into folds. No cross-validation is performed for CatBoost. Then for each unit $i$ in the inference set, we use the trained models to infer $\hat{Q}_{T_0}(x_i) = \hat{Q}_{T_0}^{-j}(x_i)$. Then these are inserted into the plug-in estimators to compute the average treatment effect, $\tau$ for each estimator below.

**Causal estimators.** After training base learners with cross-fitting, we implement the following plug-in causal estimators: backdoor adjustment (outcome regression) (Q), inverse propensity of treatment weighting (IPTW), adjusted inverse propensity of treatment weighting (AIPTW) (Robins et al., 1994).

$$\hat{\tau}_Q := \frac{1}{n} \sum_i \left( \hat{Q}_{T_1}(x_i) - \hat{Q}_{T_0}(x_i) \right) \tag{9}$$

$$\hat{\tau}_{\text{IPTW}} := \frac{1}{n} \sum_i \left( \frac{y_i t_i}{\hat{g}(x_i)} - \frac{y_i(1 - t_i)}{1 - \hat{g}(x_i)} \right) \tag{10}$$

$$\hat{\tau}_{\text{AIPTW}} := \frac{1}{n} \sum_i \left( \hat{Q}_{T_1}(x_i) - \hat{Q}_{T_0}(x_i) + t_i \frac{y_i - \hat{Q}_{T_1}(x_i)}{\hat{g}(x_i)} \right.$$
$$\left. - (1 - t_i) \frac{y_i - \hat{Q}_{T_0}(x_i)}{1 - \hat{g}(x_i)} \right) \tag{11}$$

We also use DoubleML (Chernozhukov et al., 2018) which applies an ordinary least squares on residuals from the base learners

$$\hat{\tau}_{\text{DML}} := \hat{E}\left[ (y - \hat{Q}_X(x)) \big| (t - \hat{g}(x)) \right] \tag{12}$$

# G Proof of concept pipeline for an additional subpopulation

As an additional proof of concept, we follow the same steps for the proof of concept in Section 4 but we use a different subpopulation—*Subpopulation B* for which the covariates $C$ are Engineering and Business document categories. Table 5 provides the descriptive statistics for this subpopulation. To measure the predictive accuracy for this subpopulation, we again model $P(C|X)$ with a logistic regression classifier. Averaged across held-out test folds, the F1 score is 0.92 and the average precision is 0.97.

Addressing consideration #4, we also create diagnostic plots for Subpopulation B in Figure 4. In the subsequent pipeline, we use $P^*(T|C)$ in equation 4 parameterized by $\zeta_0 = 0.85, \zeta_1 = 0.15$. Examing the modeling results in Table 6, we hypothesize that the slight performance improvement in estimators using catboost over adjustment with the oracle set $C$ is due to inclusion of extra covariates in $X$ granting additional statistical efficiency (Rotnitzky & Smucler, 2020). Regarding class imbalance, Subpopulation B is slightly more balanced (compared to Subpopulation A) with 55% business, $\mathbb{E}[Y] = 0.05$ so that smallest category ($C = 0$, $Y = 1$) has 49 documents. However, this subpopulation also suffers from finite data and class imbalance issues, as evidenced by the low average precision for the inference folds of the outcome models.

| RCT Dataset | $C$ categories | n | RCT ATE | $OR(C, Y)$ |
|---|---|---|---|---|
| Subpopulation B | Engineering, Business | 2,238 | 0.075 | 1.4 |

Table 5: For **Subpopulation B**, RCT dataset descriptive statistics including the number of units in the subpopulation ($n$) and the odds ratio, $OR(C, Y)$.

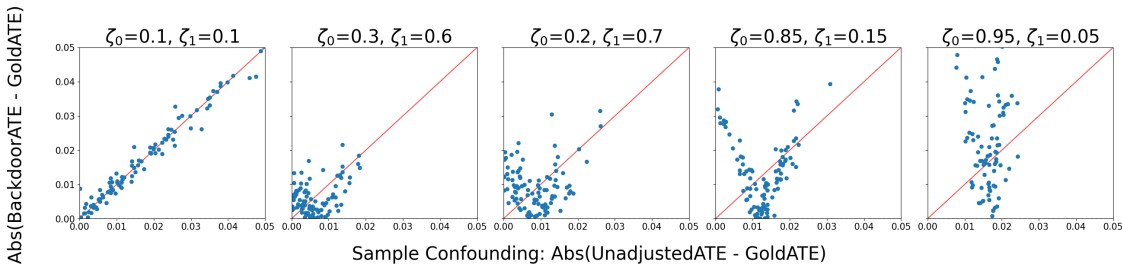

Figure 4: **Diagnostic plot for Subpopulation B.** Each plot is the parameterization of the researcher-specified confounding functions, $P^*(T|C)$ in Equation 4. Each blue dot is a different random seed (100 seeds total per plot/parameterization).

| **Prediction Ave. Prec.** ($\uparrow$ better) | $\hat{g}(x)$ train | inference | $\hat{Q}_{T_0}(x)$ train | inference | $\hat{Q}_{T_1}(x)$ train | inference | $\hat{Q}_X(x)$ train | inference |
|---|---|---|---|---|---|---|---|---|
| linear | 0.98 (0.01) | 0.78 (0.02) | 0.67 (0.24) | 0.03 (0.02) | 0.56 (0.22) | 0.17 (0.03) | 0.59 (0.17) | 0.14 (0.02) |
| catboost (nonlinear) | 0.99 (0.0) | 0.79 (0.02) | 0.97 (0.02) | 0.03 (0.01) | 0.99 (0.01) | 0.21 (0.04) | 0.99 (0.0) | 0.15 (0.03) |

| **Causal Rel. Abs. Error** ($\downarrow$ better) | Unadjusted | Backdoor $C$ | $\hat{\tau}_Q$ | $\hat{\tau}_{IPTW}$ | $\hat{\tau}_{AIPTW}$ | $\hat{\tau}_{DML}$ |
|---|---|---|---|---|---|---|
| linear | 0.14 (0.06) | 0.14 (0.1) | 2.84 (1.43) | 0.43 (0.94) | 1.51 (2.28) | 0.48 (0.2) |
| catboost (nonlinear) | 0.14 (0.06) | 0.14 (0.1) | 0.11 (0.08) | 0.11 (0.08) | 0.12 (0.09) | 0.14 (0.1) |

Table 6: Modeling results for Subpopulation B. **Top:** Predictive models' average precision (ave. prec.) for training (yellow) and inference (green) data splits. **Bottom:** Causal estimation models' relative absolute error (rel. abs. error) between the models' estimated ATE and the RCT ATE. Here, darker shades of red indicate worse causal estimates. Baselines, unadjusted conditional mean on the samples (unadjusted) and the backdoor adjustment with the oracle $C$ (backdoor C), are uncolored. We use two baselearner settings: linear and catboost (nonlinear). We report both the average and standard deviation (in parentheses) over 100 random seeds during sampling. All settings use $P^*(T|C)$ in equation 4 parameterized by $\zeta_0 = 0.85, \zeta_1 = 0.15$.

# H   Confidence Interval Plots

In Figure 5, we plot the confidence intervals given by the bootstrap percentile method (see Section 3.5 for more details). First, we construct confidence intervals for the original RCT data with a difference in means estimator. Then we plot the confidence intervals for the two sampling procedures—RCT rejection sampling and Algorithm 2 from Gentzel et al.—applied to that same RCT data. We examine synthetic DGP #1 (see Section C for details on the synthetic DGPs) for a single random seed across two different sizes of synthetic RCT data, 100K samples and 3K samples.

In the plots, the dot is the mean of the 1000 bootstrap samples. The horizontal bars indicate the endpoints of the 95% confidence interval. We note that the confidence intervals for the sampling procedures are wider because sampling reduces the size of the dataset (roughly by half). Also as expected, all the confidence intervals are wider for the smaller (3K) RCT data setting. Note, in both settings, our RCT rejection sampling contains the true (RCT) ACE (red line in the plot) while Gentzel et. al's algorithm does not.

3cm

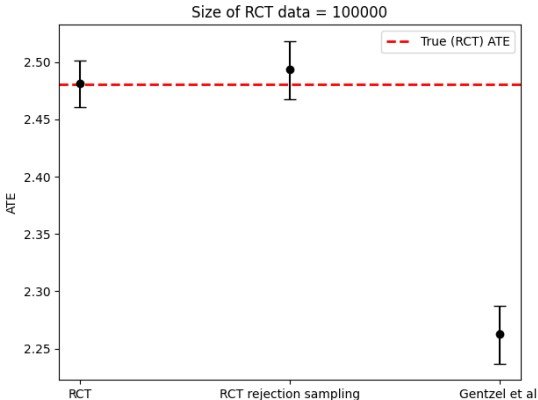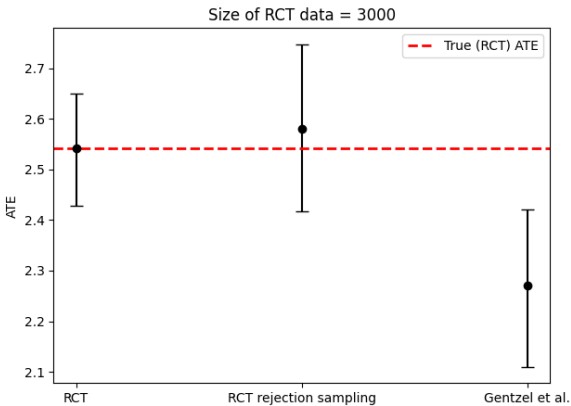

Figure 5: For Synthetic DGP #1 (single random seed) and 1000 bootstrap samples, we plot the 95% confidence intervals for the original RCT (difference in means estimator), RCT rejection sampling with a parametric adjustment (with knowledge of the oracle adjustment), and Algorithm 2 from Gentzel et al. (2021) with a parametric adjustment (with knowledge of the oracle adjustment). The mean of the bootstrap samples is denoted by the dot.

# I    Varying Confounding Strength

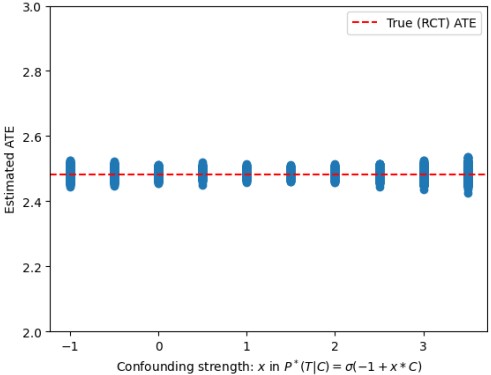

Figure 6: For Synthetic DGP #1, we alter the confounding strength $x$ in the function we specify, $P^*(T = 1|C) = \sigma(-1 + x \cdot C)$. We choose the lower and upper limits of $x$ such that $0.05 < P^*(T = 1|C) < 0.95$ to ensure *overlap* is satisfied. For each level of confounding strength, we run the rejection sampling algorithm with 1000 different random seeds. We plot the estimated ATE for the observational (confounded) dataset that is created after RCT rejection sampling. We find there are no substantial differences in the estimates due to changing the confounding strength.

