# OpenReview forum: "RCT Rejection Sampling for Causal Estimation Evaluation"
_TMLR — Accepted by TMLR_

### Review · Reviewer_Q5Ep · 2023-09-17

**Summary Of Contributions:**

This paper addresses the challenge of confounding in causal estimation from observational data, particularly in settings with high-dimensional covariates. The authors propose a new sampling algorithm called RCT rejection sampling and provide theoretical guarantees for causal identification in observational data. They demonstrate the effectiveness of their algorithm using synthetic data and present a proof-of-concept evaluation pipeline with a real-world RCT dataset. The contributions of this work aim to improve empirical evaluation for causal estimation.

**Audience:**

Yes

**Broader Impact Concerns:**

The authors have presented a Broader Impact Statement.

**Claims And Evidence:**

Yes

**Requested Changes:**

Please refer to weakness 1.

**Strengths And Weaknesses:**

Strengths:
- The paper highlights considerations when working with finite real-world data and provides a real-world example of how to put the theory and considerations into practice.

- The experimental results provided in the paper showcase the efficiency and validity of the proposed method.

- The paper is well-organized.


Weaknesses:
- The paper primarily focuses on settings where the probability of treatment is equal to the probability of control, and it may not generalize well outside of this setting. More discussion or experiments are required to demonstrate the sensitivity of the estimation results beyond this setting.

- The paper acknowledges that not all settings require empirical evaluation of causal estimators, suggesting that the applicability of the proposed methods may be limited in certain scenarios.

---

> ### Author Response · Authors · 2023-10-03
> **Response to Reviewer Q5Ep**
>
> **The paper primarily focuses on settings where the probability of treatment is equal to the probability of control, and it may not generalize well outside of this setting. More discussion or experiments are required to demonstrate the sensitivity of the estimation results beyond this setting.**
>
> We believe there has been a misunderstanding in that our contributions are the opposite of this statement. We show that previous work, Gentzel et. al 2021 from ICML, is only unbiased for the setting where the probability of treatment equals the probability of control (Table 3; second row). However, our RCT rejection sampling algorithm generalizes beyond this setting of probability of treatment equals probability of control.
>
> In Table 2 for Settings 1 and 3, the probability of binary treatment is both 0.3. See Appendix C for more details of our synthetic set-up. Under these settings, our algorithm has very low relative absolute bias whereas Gentzel et al. 2021 has high bias.
>
> **The paper acknowledges that not all settings require empirical evaluation of causal estimators, suggesting that the applicability of the proposed methods may be limited in certain scenarios.**
>
> We agree with the reviewer that not every scenario will be suited to use RCT subsampling. However, for the settings for which empirical evaluation is appropriate, our work has greatly expanded empirical evaluation for causal estimation.
>
> As we write in Section 5, “in the absence of theory or when theoretical assumptions do not match reality, we see empirical evaluation as a necessary, but not exclusive, part of the broader field of causal inference.” Multiple methods could potentially satisfy the same theoretical criteria—for example, consistent root-n asymptotically normal estimators—but one or a subset of these methods may be better suited to a particular applied setting. We believe this is where empirical evaluation is important.

---

### Review · Reviewer_fxVT · 2023-09-26

**Summary Of Contributions:**

The paper points to an issue with covariate-based subsampling of RCTs to provide real-world evaluations of causal inference methods. This is an important problem because evaluating causal inference methods on synthetic data can only take us so far in model real world causal effects. Sub-sampling from RCTs is a promising direction and the paper points out an important issue with naive covariate-based subsampling to obtain a confounded dataset.  The main idea is to adjust the covariate shift issue that comes from selecting based on a function of the covariates. This is done by using rejection sampling using the density ratio of p(T | X)/p(T) instead of sampling only based on binary samples from p(T=1 | X).

**Audience:**

Yes

**Claims And Evidence:**

Yes

**Requested Changes:**

See questions above.

**Strengths And Weaknesses:**

The paper

1. Addresses an important problem
2. is well-written and presented in a good order of concepts
2. is well-motivated and present a simple solution with supporting experiments
3. has a discussion of all the pieces in the evaluation procedure which should be paid attention to when designing a new evaluation setup.


The authors do address one of the concerns I had that "specifying a suitable confounding function P∗(T | C) may be difficult when C is high-dimensional." The issue is that the confounder can determine the label. This issue could be addressed by setting the confounder strength to be high in figure 3. Another version of figure 3 would have been useful here where, for various confounding strengths, where you put estimated ATE on the y-axis and the gold standard ATE on the x-axis. Could the authors point to whether there is already a figure that addresses this concern or produce the new figure?

My only other main comment is a high-level one. The central issue is that the marginal distribution of the covariates when subsampling. But in many cases, the covariate set of values for an RCT are pre-selected. In that case, restricting the evaluation set to have the same marginal distribution over covariates seems counter-productive. Wouldn't it be more suitable to have a target set of covariates to evaluate on that is more like the whole population of units? Can the authors comment on this?

---

> ### Author Response · Authors · 2023-10-03
> **Response to Reviewer fxVT**
>
> **The authors do address one of the concerns I had that "specifying a suitable confounding function P∗(T | C) may be difficult when C is high-dimensional." The issue is that the confounder can determine the label. This issue could be addressed by setting the confounder strength to be high in figure 3.**
>
> We thank the reviewer for this comment. However, to clarify, Figure 3 is already under a setting in which we have a proxy strategy with a $C$ that is one-dimensional and thus can directly parametrize the confounding strength, for example, by setting $\zeta_0 = 0.95, \zeta_1 = 0.05$. We used this setting to make diagnostics simple and interpretable. One could also specify $P(T|C)$ with $C=X$ where $X$ is the high-dimensional text dataset. However, for our proof of concept, we wanted to limit the degrees of freedom we had to work with.
>
> **Another version of figure 3 would have been useful here where, for various confounding strengths, where you put estimated ATE on the y-axis and the gold standard ATE on the x-axis. Could the authors point to whether there is already a figure that addresses this concern or produce the new figure?**
>
> Could you please clarify the research question you are suggesting we answer by this approach? Note, in this suggestion, the x-axis would be constant because the gold-standard ATE is the same for each RCT.
>
> **[...] the covariate set of values for an RCT are pre-selected. In that case, restricting the evaluation set to have the same marginal distribution over covariates seems counter-productive. Wouldn't it be more suitable to have a target set of covariates to evaluate on that is more like the whole population of units? Can the authors comment on this?**
>
> We thank the reviewer for this comment. The reviewer is correct that for our RCT rejection sampler, we require $P^*(C) = P(C)$ where the former is the covariate distribution from the observational (confounded) sample and the latter is from the RCT.
>
> Can the reviewer please clarify what they meant by “counter-productive”?
>
> We see two different interpretations of the reviewer’s question. If “counter-productive” means that many real-world RCTs only collect a small number of covariates, we agree that this type of RCT may not be helpful if we want to evaluate confounding adjustment methods that work best with a large number of covariates. This reinforces our point that evaluation designers may want to prefer RCTs that already have a large number of covariates.
>
> However, if the reviewer is referring to the *external validity* of the causal estimates obtained from RCT rejection sampling and some adjustment method, this is outside the scope and purpose of RCT subsampling. RCT subsampling treats the RCT as the “gold standard” and does not attempt to recover (real-world) population estimates.
>
> We are happy to add this discussion to the paper.

---

### Review · Reviewer_3Pho · 2023-09-26

**Summary Of Contributions:**

In this paper, the authors propose a new method (RCT rejection sampling) for evaluating causal inference procedures based on data from RCTs. This builds on a long line of work, dating back to Lalonde 1986 in economics which attempts to measure how well particular causal inference methods designed under no unmeasured confounding can recover average treatment effects. The authors' analysis most directly builds on recent work by Gentzel et al. 2021.

In particular, the authors (i) establish that the RCT sampling procedure developed in Gentzel et al. 2021 yields a DGP under which the ATE of the original RCT is not identified; (ii) propose a simple reject sampling procedure that fixes this problem and yields a subsampled DGP under which under which the ATE of the original RCT is identified. The authors then provide some additional discussion of practical considerations when implementing their RCT rejection sampling procedure and apply it to analyze a real world experiment run on users of a search engine.

**Audience:**

Yes

**Broader Impact Concerns:**

I have no concerns about the ethical implications of this work.

**Claims And Evidence:**

Yes

**Requested Changes:**

See my previous discussion of the paper's weaknesses.

**Strengths And Weaknesses:**

The paper tackles an important problem -- given the plethora of new methods for deriving causal inferences based on ML algorithms, it is important to have "standardized" procedures for evaluating their performance. Basing evaluations on RCTs is a natural candidate with a celebrated tradition. It is important to build up techniques for doing such evaluations based on RCTs rigorously.

Weaknesses:

1) The paper compares the researcher's estimator in an constructed subsample against the point estimate of the ATE derived in the experimental sample. There are several things that are unclear: (i) how should the ATE of the RCT be constructed? Simple difference in means? If there is some clustering, for example, in the design of the experiment, what estimator should be used? (ii) Why is the point estimate of the ATE from the RCT treated as ground-truth? There is uncertainty surrounding that quantity that should be, in principle accounted, for since the RCT sample is just some random sample from the DGP of interest. (iii) Is the authors' intuition in (ii) that the distribution of ATE estimators across the RCT rejection sampling distribution should be somehow centered at the ATE estimate from the RCT? If so, that's a result that would need to be shown.

2) The illustration of RCT rejection sampling focuses entirely on reporting the average error of the ATE estimator over the RCT rejection sampling distribution relative to the point estimate of the ATE from the RCT. Putting aside the issues in (1), why should evaluations only be focused on the MSE? Surely we care about other quantities as well -- e.g., when RCTs are run, we may be interested in testing particular effect sizes. When comparing across methods, I may care about the length of confidence intervals etc. It would be useful for the authors to discuss whether RCT rejection sampling can help inform these other desiderata.

3) The discussion of practical considerations of RCT rejection samplings are tailored to the particular application. The paper would benefit by having a streamlined presentation of how they applied RCT rejection sampling in the particular experiment they studied, and a separate "practical considerations" section that tries to speak more generally across different experimental set-ups. Surely the tradeoffs in modeling, diagnostics, and specification of P(T | C) might differ in say, the experiments analyzed in Eckles & Bakshy 2021 or more typical job training like RCTs in economics.

---

> ### Author Response · Authors · 2023-10-03
> **Response (1 of 2) to Reviewer 3Pho**
>
> **How should the ATE of the RCT be constructed? Simple difference in means? If there is some clustering, for example, in the design of the experiment, what estimator should be used?**
>
> We thank the reviewer for this comment. For our empirical proof of concept, we describe our choice in Appendix D: “The final ATE was 0.113 as computed by a simple difference in conditional means” which is an unbiased estimate of the ATE for classical RCTs.
>
> There is a tradeoff between the simplicity of this estimator and statistical efficiency (i.e., asymptotic variance) of these estimates. Here, we chose an estimator with simplicity, since statistical efficiency was not a major concern given our large sample size. Alternatively, if statistical efficiency is a concern, one could apply an estimator for the backdoor adjustment formula that adjusts for all direct causes of the outcome that are not also descendants of the treatment. However, if the outcome regression model is misspecified, this could lead to biased estimates. A middle ground is to use an AIPW estimator using an estimate of the treatment probability $P(T)$ – in an RCT, this probability is a simple proportion of treated vs. untreated – and an outcome regression model including causes of the outcome/prognostic factors. This gives improved statistical efficiency while yielding low bias.
>
> Re: the question on cluster randomized trials. The method is applicable to these scenarios as well. However, note that in a cluster randomized trial, the target parameters may vary. For example, estimators are often developed for the (weighted) sums or average of potential outcomes under a specific treatment assignment within a cluster and across clusters. That is, for a cluster $j$ with $N_j$ individuals, we may be interested in the weighted sum of the aggregated potential outcome $Y_{j}(t) \equiv \sum_{i=1}^{N_j}\alpha_{ij}Y_{ij}(t)$, or the weighted mean of the aggregated potential outcome across $J$ clusters $Y{(t)}\equiv \frac{1}{J}\sum_{j=1}^J Y_j(t)$. One would then apply the corresponding estimators that have been developed specifically for such targets – e.g., those in https://arxiv.org/pdf/2110.09633.pdf.
>
> We would be happy to add a discussion on both these points to the paper.
>
> **Why is the point estimate of the ATE from the RCT treated as ground-truth? There is uncertainty surrounding that quantity that should be, in principle accounted, for since the RCT sample is just some random sample from the DGP of interest.**
>
> We agree that uncertainty quantification is important. The confidence intervals can be easily obtained by running a non-parametric bootstrap. Please see our new CI results and plots in the updated manuscript in blue font.
>
> In the provided new plots (Appendix Section H, Figure 5), we see that the confidence intervals from the RCT and the confounded sample obtained using RCT rejection sampling procedure both contain the true ATE.
>
> **(iii) Is the authors' intuition in (ii) that the distribution of ATE estimators across the RCT rejection sampling distribution should be somehow centered at the ATE estimate from the RCT? If so, that's a result that would need to be shown.**
>
> This is correct, and follows directly from Theorem 3.2. Since $P^*$ satisfies conditions (I) and (II), we know that $\text{ATE} = g(P(C, T, Y)) = h(P^*(C, T, Y))$, where $g$ and $h$ are identifying functionals for the ATE in the RCT and rejection sampled distributions respectively. Let the true value of the ATE be $\beta$. Then, estimates obtained using any consistent and asymptotically normal (CAN) estimator – e.g. those obtained via parametric g-computation, AIPW, or double ML –  for the observed data parameters $g$ and $h$ will both be centered around the true ATE $\beta$ (a standard reference is Asymptotic Statistics, Van der Vaart 2000).
>
> We would be happy to add this clarification to the paper.
>
> **Why should evaluations only be focused on the MSE? Surely we care about other quantities as well -- e.g., when RCTs are run, we may be interested in testing particular effect sizes.**
>
> To clarify, we are not using mean-squared error (MSE) as a metric anywhere in our paper. But for the empirical proof-of-concept results (Table 3), we do measure the “relative absolute error” between a model’s estimated ATE and the RCT ATE.
>
> Note, because the rejection sampler draws from the full joint distribution $P^*(C, T, Y)$, any performance metric of interest that can be derived from this joint distribution can be used.
>
> We are happy to add this discussion to the paper.
>
> We are not quite sure what the reviewer meant by “when RCTs are run, we may be interested in testing particular effect sizes.” Can you please clarify? Thank you.
>
> **When comparing across methods, I may care about the length of confidence intervals etc.**
>
> We agree! Please see our new experiments that show how confidence intervals can be calculated using RCT rejection sampling and bootstrapping.

---

> ### Author Response · Authors · 2023-10-03
> **Response (2 of 2) to Reviewer 3Pho**
>
> **The discussion of practical considerations of RCT rejection samplings are tailored to the particular application. The paper would benefit by having a streamlined presentation of how they applied RCT rejection sampling in the particular experiment they studied, and a separate "practical considerations" section that tries to speak more generally across different experimental set-ups.**
>
> Thank you for this opinion on the structure of the writing.
>
> In Section 4, we chose to have each subsection describe a general consideration and then we provided our specific decisions for our proof-of-concept within a sub-subsection. We feel strongly that “interweaving” the general and the specific will help future evaluation designers as they create their own experimental set-ups.
>
> **Surely the tradeoffs in modeling, diagnostics, and specification of P(T | C) might differ in say, the experiments analyzed in Eckles & Bakshy 2021 or more typical job training like RCTs in economics.**
>
> We agree with the reviewer that the approach and tradeoffs for other RCTs will most likely differ from our proof-of-concept. As we say in Section 4, “Our goal is to surface questions that must be asked and answered in creating useful and high-quality causal evaluation.” We also write, “Although our approaches are specific to our proof of concept dataset, we believe other evaluation designers will benefit from a real-world example of how to put the theory and considerations into practice.”
>
> Both Eckles & Bakshy 2021 and Lalonde 1986 approached empirical evaluation of causal estimates from a constructed observational study framework, not an RCT subsampling framework (see Section 2). However, we agree with the reviewer’s (implicit) suggestions that we could (hypothetically) use the RCTs from these two studies in RCT subsampling.
>
> Lalonde 1986’s RCT only has four covariates (taken from Lalonde 1986 Table 4, footnote c:
> age, years of schooling, high school drop-out status, race). It’s possible that several of these covariates would not correlate with $Y$ (Section 4.2 in our paper, Consideration #1) and there would only be 1 or 2 covariates available for the remainder of the experimental pipeline. In this scenario, the benefit of empirical evaluation of adjustment methods may be limited. As we write in our introduction, “In settings with only a few covariates, simple estimation strategies—e.g., parametric models or contingency tables—often suffice to compute the adjusted estimates.”
>
> Regarding Eckles & Bakshy 2021, their Facebook RCT is proprietary and thus not publicly available. However, if we were to hypothetically use the RCT, we would have access to over 3700 covariates, the vast majority of these covariates being the number of times a specific URL was shared on Facebook in a six month period. It’s likely with this many covariates Consideration #1 (Section 4.2) would hold: that many of these covariates would be correlated with the outcome, $Y$.
>
> Using many correlated covariates, there would be a large number of degrees of freedom for setting $P*(T|C)$, which could be beneficial for evaluation. For example, one could design evaluations to understand *dilution* of the confounders---only a subset of $C$ influence $T$---or manipulating the degree of influence of various $C$, or make the relationship between $T$ and $C$ being linear or non-linear.
>
> We are happy to add this discussion to the paper.

---

### Author Response · Authors · 2023-10-03
**Overall response to all reviewers**

We thank all three reviewers for reading our paper and providing helpful comments. Particularly, we thank you for highlighting the importance of this work:
- Reviewer 3Pho: “The paper tackles an important problem -- given the plethora of new methods for deriving causal inferences based on ML algorithms, it is important to have ‘standardized’ procedures for evaluating their performance.”
- Reviewer fxVT said our work “addresses an important problem” and “is well-motivated and present a simple solution with supporting experiments”
- Reviewer Q5Ep: “The experimental results provided in the paper showcase the efficiency and validity of the proposed method.”

Following reviewers’ suggestions, we ran several new experiments about confidence intervals (CI) and their coverage:
- CI coverage results for the synthetic DGPs in Table 2, and
- A new appendix section (Section H) with CI plots

We have incorporated these experiments and text supporting those changes into the manuscript and we use blue font to indicate what is new.

**Tip on Open Review platform**: Now, when you click the "PDF" logo in the upper right hand corner of this page, you will see our updated manuscript with the revisions.

---

### Decision · Action_Editor_YvoY · 2023-11-02

**Recommendation:** Accept as is

**Comment:**

The paper is well-written, both in terms of technical details and the overall structure and presentation. As mentioned above, the main claims are precise, closely linked to substantive results presented in the paper, and are not in my opinion overstated. Their novel proposed procedure builds on the existing RCT sub-sampling research in a natural way, and all the reviewers were in favor of acceptance following discussion with the authors. I agree, and feel that the paper can be accepted as-is.

**Audience:**

The intersection of causal inference and machine learning methods is a burgeoning area. This paper provides practical procedures for evaluating causal effect estimators with formal guarantees of certain advantages over existing related methods, plus empirical analysis with code and data to be made public. I feel confident the paper will have a sufficient audience.

**Claims And Evidence:**

The main claims of this paper are related to procedures for evaluating estimators of causal effects. Their starting point is the well-established literature on randomized controlled trial (RCT) sub-sampling methods. Formally, they highlight issues with existing methods by showing that it is possible for such methods to sample in a fashion which stymies the ultimate identification of causal effects. They show how an augmented method (RCT "rejection" sampling) can remove this limitation, and put effort info empirical tests that highlight both the potential for improvement suggested by the theory, and work as a sort of boilerplate for a general-purpose workflow centered around their proposed method. The main claims are tightly bound to the concrete results presented in a well-structured narrative, and to the best of my judgement based on my reading of the paper and the reviews, the claims are solid.